Multi-user conflict resolution mechanisms for smart home environments

http://orcid.org/0000-0003-2003-6310 Aljawarneh Mahmoud Mohammad 1 2
http://orcid.org/0000-0002-0953-4055 Shah Shahid Munir 3 shahidmunirshah@yahoo.com
Dhomeja Lachhman Das 4
Malkani Yasir Arfat 5
Jawarneh Mahmoud Saleh 1 2
1 Faculty of Information Technology, Applied Science Private University , Amman , Jordan
2 MEU Research Unit, Middle East University , Amman , Jordan
3 Department of Computing, Hamdard University , Karachi, Sindh , Pakistan
4 Institute of Information and Communication Technology, University of Sindh , Jamshoro, Sindh , Pakistan
5 Department of Computer Science, University of Sindh , Jamshoro, Sindh , Pakistan
Winkler Robert
Electronic publication date: 2023 Jun 23
Publication date: 2023
Volume: 9
Electronic Location ID: e1443
Received 2022 Dec 28; Accepted 2023 May 25
Copyright: © 2023 Aljawarneh et al.
Copyright year: 2023
Copyright holder: Aljawarneh et al.
License: This is an open access article distributed under the terms of the Creative Commons Attribution License, which permits unrestricted use, distribution, reproduction and adaptation in any medium and for any purpose provided that it is properly attributed. For attribution, the original author(s), title, publication source (PeerJ Computer Science) and either DOI or URL of the article must be cited.
License URL: https://creativecommons.org/licenses/by/4.0/

Keywords: Automatic resolution, Conflict resolution, Mediated resolution, Smart environments, Smart home, Pervasive computing, Ubiquitous computing

Funding: The authors received no funding for this work.

==============================
Context-awareness is a pervasive computing enabling technology that allows context-aware applications to respond to multiple contexts such as activity, location, temperature, and so on. When many users attempt to access the same context-aware application, user conflicts may emerge. This issue is emphasized, and a conflict resolution approach is presented to address it. Although there are other conflict resolution approaches in the literature, the one presented here is unique in that it considers the users’ special cases such as their sickness, examinations, and so on when resolving conflicts. The proposed approach is helpful when several users with different special cases try to access the same context-aware application. To demonstrate the usefulness of the proposed approach, a conflict manager is integrated with the UbiREAL simulated context-aware home environment. The integrated conflict manager resolves conflicts by taking users special cases into account and employing either automated, mediated, or hybrid conflict resolution approaches. The evaluation of the proposed approach demonstrates that users are satisfied with it and that it is critical and essential to employ users’ special cases in detecting and resolving users conflicts.

Introduction

A brief overview of pervasive computing (context-aware computing)

In 1991, Mark Weiser first proposed the concept of ubiquitous computing (Weiser, 1991), now often referred to as pervasive computing. According to his vision (which leads to smart environments), in future computing would move beyond the desktop and become widespread and invisible to individuals, in a sense that users interacts with the environment in a subconscious state. Context-awareness is the main pillar of pervasive computing that uses users’ contextual information (i.e., occupancy, activities, weather, location, etc.) and provide them with the service(s) of their interest (Abowd et al., 1999; Emmanouilidis, Koutsiamanis & Tasidou, 2013). Although context-awareness is a key component of managing daily activities in smart environments, there are a number of challenges associated with it. User control, inconsistent contexts, energy consumption, data privacy, security, and conflicts among users are a few examples (Dhyani et al., 2022; Rao & Prema, 2021; Alsamhi et al., 2022). While the research community is already investigating these challenges, more in-depth and focused research is required to broaden the scope of this fascinating area.

Smart environments and multi-user conflicts

A smart environment is made up of interconnected sensors, actuators, appliances, and applications. The connected gadgets adapt themselves according to the contexts to increase comfort and safety for their users. It is difficult to handle several users in a smart environment because it is ideally designed for a single user. Many users share a range of resources, and the environment is in charge of maintaining and structuring its resources to make them more accessible to its users. Multi-user conflicts occur when multiple users attempt to use context-aware applications at the same time or they have different priorities and preferences. For example, when user A enters a living room, the temperature and lighting control applications adjust to her preferences. But what if user B enters the same living room with a different set of lighting and temperature preferences?

Need to resolve multi-user conflicts

Devices in smart environments (specifically smart home environments) are typically controlled by a specific application embedded in the system device. This device should ideally be controlled by a single trusted person. A multi-user, multi-device smart home environment always pose different challenges like security, privacy, and manageability (Hua et al., 2022; Zeng & Roesner, 2019). Users have contradictory, complicated, and continuously changing demands on various devices, resulting in difficult-to-manage conflicts. So, in order to carry out daily tasks without difficulty in such circumstances, a thorough conflict resolution mechanism is needed (Ospan et al., 2018). Although multi-user conflict resolution has been a focus of study for decades, with an ever-growing Internet of Things (IoT) network and applications, it is necessary to focus more on meeting users’ requirements continuously.

Challenges of the traditional approaches to resolve multi-user conflicts

As discussed in “Literature review”, multiple strategies have been employed to handle multi-user conflicts. Most of these methods automatically detect and resolve user conflicts without the users’ active involvement. Nonetheless, there may occasionally be circumstances where users’ participation and discussion are needed to resolve the conflicts. User participation in conflict resolution is crucial since it improves the harmony of the home’s residents, as advocated in the literature. (Del Rio, 2022; Shin, Dey & Woo, 2010; Shin & Woo, 2009b). This calls for the environment to suggest the users with the best resolution candidates to resolve conflicts. Although these solutions successfully manage user conflicts, they fall short of dealing with some of the special cases. To the best of our knowledge: During conflict resolution, existing solutions do not take into account users’ special cases such as illness, examinations, visits, etc.

To better resolve conflicts through discussion among users (mediation), the users’ involvement should be as little as feasible; nevertheless, the present literature is deficient in addressing this issue.

If the user is not interested in using a certain application, she may be excluded from the conflict resolution discussion. This issue may lengthen the conflict resolution process; yet, the existing literature ignores this critical issue.

Research questions

Based on the research gap presented above, this research aims to address the following research questions: By factoring in user special cases into standard conflict resolution systems (such as automatic, mediated, and hybrid, see “Literature review” for more detail on these approaches), can multi-user conflicts be managed more effectively?

How is user involvement reduced during a mediation-based approach to conflict resolution?

If the user does not want to utilise a certain application, how is the conflict resolved?

Hypothesis

The null hypothesis ( H0) and the alternative hypothesis ( H1) for the respective research questions are the following: H0(1): Multi-user conflicts resolution in smart home environments is improved by including users’ spacial cases into existing automatic, mediated, and hybrid conflict resolution approaches.

H1(1): Conflict of at least one of the existing conflict resolution approaches (automatic, mediated, and hybrid) remained the same by including users’ spacial cases.

H0(2): During mediation conflict resolution approach, users’ involvement is decreased by automatically adapting to the preferences of the special case users?

H1(2): During mediation conflict resolution approach, users’ involvement of at least a single user remained the same by automatically adapting to the preferences of the special case users?

H0(3): Users’ conflicts in smart home environments is detected and resolved even if the user herself is not interested in using a particular application/service.

H1(3): Users’ conflicts in smart home environments is not detected and resolved even if the user herself is not interested in using a particular application/service.

Research goal

The purpose of this study is to suggest a method for resolving conflicts between several users in environments like smart homes by actively involving the users and taking into account their special cases, such as illness, exams, visitors, etc.

Research contributions

Following are the contributions of this research: The proposed conflict manager takes into account users’ special cases (as stated above) for determining a resolution approach to be applied to detect and resolve users’ conflicts.

To the best of our knowledge, no current literature has taken user special cases into account when choosing the resolution strategy to settle users’ conflicts, making this a novel method for resolving users’ conflicts.

Users’ conflicts are resolved utilising the automatic, mediated, and hybrid resolution mechanisms that are already in place, but now users’ special cases are incorporated.

Users’ conflicts for the special case users are resolved using mediated technique with less user engagement, and the user involvement during mediation has been reduced by letting the applications automatically adjust to the special case users’ preferences.

The situation where the user herself is not interested in the application that is offered is regarded significant and is included in the proposed system to detect and resolve multi-user conflicts. This is an important but neglected issue that is also taken into account.

The proposed system is tested and evaluated utilising usability testing, which is implemented in Java. According to the usability research, the proposed system can be used to identify many conflicting circumstances with special cases and can soothe people by quickly resolving their conflicts.

The reminder of the article is organized as follow: “Literature review” presents the review of the related works that address the multi-user conflict issues in smart environments. “Methodology” presents the methodology of the proposed study that includes research design and complete experimentation of the proposed system. Results and discussion is presented in “Results and discussion”. Finally, conclusion remarks and future directions are presented in “Conclusion and future directions”.

Literature review

This section briefly summaries efforts used to identify and resolve multi-user conflicts in smart environments. According to the studied literature, three approaches are frequently employed for identifying and resolving multi-user conflicts. Each of these strategies is detailed below.

Automatic conflict resolution approach (ACRA)

This strategy focuses on automatically resolving conflicts based on user priorities and/or preferences without users’ active participation. Various approaches have been used, using preferences in some cases, priorities in some other cases or a combination of both priorities and preferences. The method proposed in (Haya et al., 2006) suggested three algorithmic conflict-resolution techniques. The three methods are: (1) the fair principle, which is based on user preferences; (2) use first, which assigns priorities to the user who enters the environment first; and (3) preference priority, which gives priorities to preferences and settles disputes using the preference with the highest priority. The authors also advocated for giving the sickest user the highest priority during a quarrel and adjusting priorities or preferences accordingly.

Carreira & Sílvia Resendes (2014) considered conflicts resolution as the constraint satisfaction problem (CSP). Their system automatically resolved conflicts based on users’ preferences using some constraints. Constraints are valid range of values for users preferences and services that enable performing activities in environment. In case of non-satisfiable constraints, the system assists users in resolving conflicts by mediating of resolution candidates. Research work presented Camacho et al. (2014) also used CSP for conflicts resolution. The difference is that the latter used ontologies to detect and satisfy constraints imposed on the environment. Ontologies, is one of the concepts used in smart environments. It allows grouping of devices based on similarities e.g., device kind, device location, etc. (Elenius & Ingmarsson, 2004). Chaki, Bouguettaya & Mistry (2020) also formulated multi-user conflicts as ontology conflicts. Their system was able to detect whether a conflict happened in a single application or in a number of them, as well as whether it was a functional or a non functional conflict.

Chaki & Bouguettaya (2020) gathered information about users’ device and service usage patterns using the principles of entropy and information gain (IG), and then created an algorithm based on temporal proximity to identify and resolve conflicts. “Kratos” is a multi-user and multi-device aware access control mechanism that was developed by Sikder et al. (2020, 2022). The system consists of three parts: (a) a user interaction component that enables users to specify their access control settings; these settings are then translated into policies in the system’s second component; (b) a backend server; and (c) a policy manager that examines these policies to determine how to negotiate conflicts between users and creates final policies that will be used to resolve conflicts.

Mediated conflict resolution approach (MeCRA)

In certain circumstances, such as public settings and gatherings, identifying preferences and priorities may be challenging. Researchers have concentrated on settling conflicts in these circumstances by taking into account the preferences of the majority of users. Jukola (O’Hara et al., 2004) is a music mediator system for a public place i.e., cafe. It allows customers to influence the selection of songs played in the cafe. The system provides the customers a device on every table to mediate the list of songs showed on a shared display screen. After selecting songs by the customers, it plays the most rated song. This approach requires active users’ participation to resolve the conflicts.

As compared to public places (like restaurants), private places (like homes) require different kind of mediation as the home members can easily resolve their conflicts through discussion. Shin, Yoon & Woo (2007) in their research, proposed a user-centric conflict management system that considers different contexts and recommendations of a personal companion. Their system allows users to select from the recommendations of their personal companions to resolve conflicts. Mediation process enables users to exchange their opinion regarding media content to be agreed upon an item that reflects all the users’ preferences. Shin & Woo (2009a) proposed a conflict management framework for smart home environments where they used ontology to detect conflicts. An approach determination tree was used to assign an appropriate resolution approach to the detected conflicts. Based on the selected approach, conflict was resolved either by automatic decision or through discussion among the users.

Garg & Cui (2022) analysed three types of the most occurring conflicts in real life setting environments i.e., activity conflict, value conflict, and preference conflict. The conflict resolution was left on users so that they could resolve it through their mutual discussions as happens in their real life where the conflicts are either resolved by compromises or through mutual negotiations. Based on their findings, a set of considerations were provided that enable designers to design future IoT that may better fit into people’s homes and everyday lives.

Hybrid conflict resolution approach (HyCRA)

In a household setting where several conflicting circumstances could arise, resolving conflict with one resolution strategy would not lead to good results. As a result, a system supporting several techniques and multiple schemes is required to handle various conflicting circumstances.

Otto et al. (2006) designed a system that resolves conflicts using an input control device through explicit user interaction. Their solution used different approaches to deal with multi-user conflicts such as (a) giving priority to that person only who has the input control device, (b) allowing the person who entered first into the environment to have the input control device, (c) giving a specific time to every person to have the input control device, (d) involving every person to actively participate in the conflicts through mediation using personal digital assistant (PDA). In the last case, the system involved all the users even if they were not part of the conflicts and it then adapted itself according to the input to which all the users were agreed. Shin & Woo (2009b) proposed a socially aware TeleVision (TV) through which conflicts were resolved in either of two methods (a) automatically based on users profiles and, (b) by recommending users a common group profile. Their system provided a remote control that allowed users to mediate the final decision. The system then provided a final decision based on users’ recommended common group profile. The work presented in Shin, Dey & Woo (2010) resolved conflicts by either of two methods (a) using profile based automatic approach, and (b) by the use of social mediation. In social mediation approach, the users engaged in negotiating for a proper resolution. The system had a balance model to evaluate a group feelings to reduce a discussion time. Bisicchia, Forti & Brogi (2021) proposed a a declarative framework along with its opens source Prolog prototype “Solomon”, to specify policies for mediating contrasting goals and actuator settings in smart environments. Their proposed prototype resolved user to user and user to admin conflicts into a target state for the smart environment and its actuators.

To acquire a more in-depth understanding, the above-mentioned literature is also provided in a table. Table 1 gives a tabular form of the above-mentioned literature review.

Table 1 Summary of conflict resolution approaches used in literature.

S.N	Authors	Title	Approach used	Contributions	
1	Sikder et al. (2020, 2022)	Who’s Controlling My Device? Multi-User Multi-Device-Aware Access Control System for Shared Smart Home Environment.	ACRA	Multi-user, multi-device aware access control systems called “Kratos and Kratos+” were proposed.

The proposed system’s interaction module was in charge of translating the users’ desired access control settings into access control policies.

The policy manager was in charge of analysing policies and starting automated negotiations among users to settle conflicting demands.

	
2	Garg & Cui (2022)	Social Contexts, Agency, and Conflicts: Exploring Critical Aspects of Design for Future Smart Home Technologies.	MeCRA	Three types of most occurring conflicts in real life setting were discussed i.e., activity conflicts, value conflicts, and preference conflicts.

The conflict resolution was left on the users so that they can resolve it through their mutual discussions (as happens in their real lives).

	
3	Bisicchia, Forti & Brogi (2021)	Declarative Goal Mediation in Smart Environments.	HyCRA	A declarative framework along with its open source Prolog prototype “Soloman” was presented.

The purpose was to specify policies for mediating goals and actuator settings in smart environments.

The system was able to resolve user to user and user to admin conflicts.

	
4	Chaki & Bouguettaya (2020)	Fine-grained Conflict Detection of IoT Services.	ACRA	A conflict detection framework for IOT based services in multi-resident smart home environments was presented.

Conflicts were classified using Entropy and Information Gain.

A novel a-priori algorithm was designed based on temporal proximity to provide the foundation to resolve users’ conflicts.

	
5	Chaki, Bouguettaya & Mistry (2020)	A Conflict Detection Framework for IoT Services in Multi-resident Smart Homes.	ACRA	A novel framework was proposed to detect conflicts among IoT services.

Ontology (categorizing the devices according to their similarities) was used to categorize different types of conflicts.

A hybrid conflict detection algorithm was presented for conflicts resolution.

	
6	Camacho et al. (2014)	An Ontology-based Approach to Conflicts Resolution in Home and Building Automation Systems.	ACRA	An ontological framework for conflict detection and resolution backed by knowledge-based representation was proposed.

Conflict detection was accomplished using the SPARQL Protocol and RDF Query Language.

Conflict resolution was accomplished by finding the best possible combination of services and by performing constraint solving.

	
7	Carreira & Sílvia Resendes (2014)	Towards Automatic Conflict Detection in Home and Building Automation Systems.	ACRA	A constraint solving based framework to detect and automatically resolve conflicts was presented.

The system was specifically designed to resolve conflicting situations that frequently occur in home and building automation systems.

	
8	Shin, Dey & Woo (2010)	Toward Combining Automatic Resolution with Social Mediation for Resolving Multiuser Conflicts.	HyCRA	A hybrid resolution mechanism that combines social engagement and automatic resolution was developed.

Different contexts like preferences, priorities, and types of applications were used to select an appropriate resolution method for the encountered conflict.

	
9	Shin and Woo (2009a)	Service Conflict Management Framework for Multi-user Inhabited Smart Home.	MeCRA	A conflict management framework was presented in which ontologies were used to detect conflicts.

An approach determination tree was used to assign an appropriate resolution approach to the detected conflict.

Based on the selected approach, conflicts were either resolved automatically or through discussion among users.

	
10	Shin, Yoon & Woo (2007)	Media Service Mediation Supporting Resident’s Collaboration in ubiTV.	MeCRA	A context-based mediation method, consisting of service mediators and mobile mediators was proposed.

The service mediators were used to detect service conflicts and recommend their preferred media contents on mobile devices.

The mobile mediators were used to collect the recommendations and give the users personal recommendation.

With combination of the service and mobile mediator, the residents were allowed to negotiate the media contents.

	
11	Otto et al. (2006)	A User Survey on: How to Deal with Conflicts Resulting from Individual Input Devices in Context-Aware Environments.	HyCRA	Four simple approaches were used for conflict resolution i.e., (1) giving priority to that person who has the input control device, (2) allowing the person who entered first into the environment to have the input control device, (3) giving a specific time to every person to have the input control device, (4) involving every person to actively participate in the conflicts through mediation using personal digital assistant.

	
12	Jinghua & Wolfgang (2005)	Profile Management Technology for Smart Customization in Private Home Applications.	ACRA	A profile management framework was presented for situation-dependent customization in smart home environments.

Three different strategies were presented i.e., Fair Principle, Use First, and Preference Priority to automatically revolve users’ conflicts.

	
13	O’Hara et al. (2004)	Jukola: Democratic Music Choice in a Public Space.	MeCRA	An interactive MP3 device i.e., Jukola was designed to allow group of people in a public place (i.e., cafe) to democratically choose the music.

A public display was used to nominate songs which were subsequently voted on by people using networked wireless handheld devices.

The most rated song was played.

	

Summary of literature review

Literature review presented above suggests that a multitude of research has been conducted in the proposed research area, resulting in proposition of different resolution algorithms (based on priorities and/or preferences) and approaches (ACRA, MeCRA, and HyCRA) to detect and resolve multi-user conflicts. Some of the proposed algorithms are suitable for public places (e.g., restaurants), while others are suitable for the private places (e.g., homes). The importance of the MeCRA in private places is stressed in the reviewed literature but the context-aware home environment dictates the need of minimizing users involvement to reduce their distractions, especially in situations where users may have special cases (i.e., illness, preparation for examination and guests). In the existing systems, if ACRA is applied for such special cases, it may lead to unpleasant results for those special case users. Moreover, if the MeCRA is applied, this will lead to a discussion among home users and the result will most likely be the home users conceding their right to the special case users as a resolution for the conflicts to provide them comforts. Since, the users’ special cases are temporary situations that might occur for specific amount of time. Through mediation, the family members show care for each other by conceding everyone’s rights of using the applications especially the one who has special cases (i.e., illness). This allows the family to live in more harmonic situations by providing the special case users the feelings that the other home members are caring for them. However, to better resolve the conflicts, there is a need to decrease the users involvement during mediation.

In order to lessen the users involvement, mediation can be minimized by allowing the applications to automatically adapt to the preferences of the special case users. The same has been focused here in this research. Also, an important but neglected aspect is considered i.e., in case if user herself is not interested in the application available at the vicinity such as TV (maybe because of work overburden or some other reasons). This situation is considered important and embed in our proposed approach to multi-user conflict detection and resolution. While the HyCRA to multi-user conflict detection and resolution has been used in the literature in which some conflicts are resolved using ACRA and others using MeCRA, none of the existing systems have considered use of special cases in the decision making of the selection of the resolution approach. The proposed approach takes into account special cases in determining a resolution approach to be applied to detect and resolve the multi-user conflicts in the smart home environment.

Methodology

The research design flow diagram for the proposed study is shown in Fig. 1. The proposed study’s research design, as shown in Fig. 1, entails the following steps:

Figure 1 Research design of the proposed system.

1) Designing high-level architecture.

2) Participants selection to participate in the study.

3) Environment selection for conducting experiments.

4) System implementation.

5) System testing.

6) System evaluation.

This is a description of each stage in detail:

High level architecture of the proposed system

Figure 2 depicts the proposed system’s high-level design. UbiREAL simulator, users’ profiles, and users’ conflicts manager make up its three primary parts. This is a brief explanation of how each of these components function.

Figure 2 High-level architecture of the proposed system.

UbiREAL simulator

The UbiREAL simulator has built-in simulations of sensors, actuators, and applications (for more information on the UbiREAL simulator, see “Environment selection”). Devices, users’ interactions with them, and the users’ movements within the simulated home surroundings are all detected by sensors. Applications are in charge of providing information about the names of devices and the actions that can be carried out on them.

Users’ profiles component

The users’ profiles component is in charge of keeping track of user profiles. Each profile includes data that should be taken into account when settling conflicts. For example, the user’s name, her priorities, her preferred methods of using various applications, and any special cases (if exist).

Conflict manager component

The conflict manager component is responsible for detecting and resolving multi-user conflicts. Its working is assisted by three sub-components i.e., conflict detection component, determination approach component, and resolution component.

(1) Conflict detection component: Its goal is to identify conflicts, collect information about them (such as involved users’ identities, profiles, and the location of the conflict, etc.), and deliver that information to the determination approach component to select the best strategy for resolving them.

(2) Determination approach component: Based on the data from the conflict detection component, it is incharge of choosing an effective resolution strategy. Its operation is aided by an algorithm i.e., “approach determination structure” (see Fig. 3), which helps choose the best conflict resolution strategy from among ACRA, MeCRA, and HyCRA (see “Literature review” for more information on these strategies).

Figure 3 Approach determination structure.

As shown in Fig. 3, the approach determination structure selects ACRA in the following four cases: 1. If there is no special case user involved in the conflicting situation.

2. If there is only one special case user from the involved users.

3. If there are multiple special case users and the deviation in their preferences is low.

4. If there are multiple special case users and the deviation in their preferences is high.

MeCRA is selected when the involved users have the same special case and the deviation in their preferences is high. Finally, HyCRA is selected when there are multiple special case users with the multiple applications present in the environment and ACRA is not applicable. Once the appropriate resolution approach is selected, it is then sent to the resolution component to resolve the conflict.

(3) Resolution component: This component is responsible for making resolution about the detected conflict based on the selected resolution approach. The conflict might be resolved automatically without active users’ involvement, or by mediating some resolution candidates based on the involved users’ preferences, letting the users discuss among themselves and selecting the appropriate resolution approach. The conflict manager, after resolving the conflict, passes the values to the application(s), which adapts itself/themselves according to these values.

Participants selection

A total of 84 people were chosen from the academic community to take part in the usability study. Table 2 displays the demographic data for the participants, who were a mix of students and teachers. The participants were divided into 21 groups, each with four participants. Furthermore, each participant of each group was assigned a role in the scenario that was similar to their actual role in the family.

Table 2 Participants selection criteria.

Gender	
Male	79%	
Female	21%	
Age	
Less than 18	0%	
18–25	87%	
26–35	11%	
36 or Above	2%	
Role in the family	
Father	4%	
Mother	0%	
Children	96%	
Education	
High school college	13%	
Bachelor	67%	
Masters	19%	

All methods/experiments were carried out in accordance with relevant guidelines and regulations as well as all experimental protocols were approved by Ethics Committee of University of Sindh, Jamshoro. Ethics approval and participant consent was taken as per policy of the University of Sindh, Jamshoro. All subjects in the database were enrolled at the university and have given informed consent, and if under 18, consent was taken from parent and/or legal guardian. Additionally, all the subjects have given the right to withdraw form the study at any time. Furthermore, an informed consent was taken from all subjects and/or their legal guardian(s) for publication of identifying information/images.

Environment selection

In order to implement the study a simulated virtual environment i.e., UbiREAL (Nishikawa et al., 2006; Alshammari et al., 2017) was selected. UbiREAL is a three dimensional (3D) virtual environment that provides a suitable environment to test the context-aware applications and allows to visualise the state change of devices through a 3D Graphical User Interface (GUI). UbiREAL simulator was made public with the source code in the 2012.

Implementation

The proposed method has been implemented on UbiREAL using Java. To detect and address conflicts among users, the proposed conflict manager with the UbiREAL simulator is built on top of UPnP (see Fig. 4). The conflict manager, which is also a UPnP client control point, subscribes to sensor events to learn when a device’s state changes. Using this information, along with user profiles (which were created using XML, as shown in Fig. 5), a resolution is then suggested, which ends up resolving the conflicts. To edit the XML files according to the users’ preferences, a GUI was created. The straightforward XML editor in Fig. 6 just comprises the Open and Quit buttons. When the open button is pressed, a new dialogue box allowing for the selection of an XML file for editing displays. After that, users may specify their names and preferences through a GUI without having to manually change an XML file. Another Java-based GUI was created to allow users to choose a resolution from the suggested resolution candidates when a disagreement arose. The Java-based GUI for choosing a resolution from among the suggested resolution candidates is displayed in Fig. 7.

Figure 4 Structure of UbiREAL simulator.

Figure 5 XML-based user profile sample.

Figure 6 GUI for XML-based user profile editor.

Figure 7 GUI for the recommendations of resolution candidates based on the involved users’ profiles.

System testing

The proposed approach was tested through the usability study of the implemented system. The test experiments were carried out in a room with a projector that projected the UbiREAL smart environment in front of the participants (see Fig. 8). The participants were told to interact with the environment from the front so that they might feel as though they were inside the environment. Before the usability study began, the participants were given a briefing on the users’ conflicts in smart environments and the methods employed to resolve them automatically. They were also briefed about the necessity of taking users’ special cases into account when resolving conflicts as well as the working of the proposed system in resolving users’ conflicts by taking users’ special cases into account.

Figure 8 Users interaction with the simulated environment.

The usability study’s participants were split up into 21 groups, each of which had four individuals. Each participant from each group received a role to play in accordance with their family roles (i.e., parents were given a parent role; children were given a child role etc.). Two distinct scenarios were created based on roles to be executed during the experiments (for more information on the constructed scenarios, see “Evaluation”). The scenarios were designed to test ACRA, MeCRA and HyCRA resolution approaches. Using these approaches and based on users’ preferences, a resolution candidate was recommended by the proposed system. Based on the recommended resolution candidate and discussion among users, an actor was appointed to apply the selected resolution candidate to resolve the conflicts. Using projected smart environment, the usability was performed as follows:

Initially, in case of recommendations, the recommendations were popped up on the screen of the projected smart environment. A GUI based utility was added to set the preferences of users (see Fig. 9). The settled users’ preferences were saved as a separate profile for every participant. After the users’ preferences have been saved, the experimental situation involving the applications and the degree of deviations in the users’ preferences were controlled. Three applications were selected and projected on the wall screen by the simulation i.e., (1) simulated air conditioner application (2) simulated television application, and (3) simulated light application (see Figs. 10A–10C). As the conflict occurred, the conflict manager detected the conflict and responded immediately to the system to employ the ACRA or MeCRA for the resolution.

Figure 9 GUI utility for user’s preference setting.

Figure 10 Projected (A) air conditioner (B) TV screen (C) light appliance.

Evaluation

To evaluate the performance of the proposed system with multiple applications, multiple users and different special cases, two different scenarios were executed. Considering multi-user conflicting home environment and different special cases, in these scenarios, every participant was given a role and a special case.

Scenario-I

The first scenario was executed with two family members and their two friends. The first user entered the environment was a home member and having an illness special case. The second user entered the environment was a friend of the first user who came to visit him at his home. The second user has a guest special case. The third user was the sister (home member) of the first user, and she has an examination preparation special case. The fourth user was a friend of the second home member with a guest special case but at that instance of time she was not interested in using the applications in the environment.

Scenario-II

The second scenario was executed with four home members: a father, a daughter, and two sons. The first user entered the environment was a son with no any special case (normal). The second user entered the environment was the brother of the first user with an illness special case. The third user entered the environment was the father who was not interested in using any application running in the environment. The last user entered the environment was the daughter with the examination preparation special case.

In order to examine the system’s behavior of selecting the appropriate resolution approach based on users’ special cases and the degree of deviations in their preferences, very scenario was executed twice: the first time was with a low deviation in the users’ preferences, while the second time with a high deviation in the users’ preferences. The reason of using only two scenarios in the usability study was to avoid users’ exhaustion because every single scenario took around fifteen minutes to complete. Executing only two scenarios allowed us to get a reasonably realistic data without taking much time of the participants.

The following three surveys were distributed to the participants in order to gauge their opinions of the proposed system and how well it handled user conflicts during executed scenarios: 1. Pre-Test Questionnaire (PRTQ): PRTQ was distributed to gather users’ demographic information.

2. After Scenario Questionnaire (ASQ): ASQ was distributed to collect users’ opinion for different aspects of the proposed system.

3. Post-Test Questionnaire (POTQ): POTQ was distributed to gather users’ feedback and suggestions about the broader aspects of the proposed system in resolving multi-user conflicts.

In the beginning of the experiments, PRTQ was distributed among the participants to fill their demographic information. During the experiments, some of the resolutions were selected by the conflict manager using mediated resolution approach, which recommended the involved users some resolution candidates and the users had to discuss and select one of the candidates. Then the system adapted according to that selection. After completing the scenario each participant was given ASQ to gather their satisfaction about different aspects of the proposed system. Finally, after completing the test experiments, every participant was given a POTQ that contained only two questions. One about the approach, which was adapted by the conflict manager in case there was no any special case user, and the other was about the rating of different resolution approaches, which the users experienced while executing the scenarios.

Results and discussion

In this section, results obtained from the usability study and the analysis of the system features are presented. Usability analysis of the proposed system provides an assurance that the system is easy to use and the intended users are satisfied with its working in detecting and resolving multi-user conflicts. Data obtained from both ASQs and POTQs reflects the participants’ opinion about the overall usability of the proposed system as well as its efficiency in selecting the appropriate resolution approach for conflict resolution as it occurred in the scenarios. Satisfaction of users with the proposed system is expressed in term of rating score on 1–7 scale. Selection of seven-step scale is based on (Lewis, 1995; Bates & Bierton, 2000) that captures the best discrimination of users. In the seven-steps scale of the proposed system, the score 1 is the lowest performance (unsatisfactory performance) indicator, while the score 7 is the highest performance (the most satisfactory performance) indicator. The graph shown in Fig. 11 is drawn from ASQs and POTQs submitted by the participants of the usability study that provides the summary of the overall average users satisfaction scores about different aspects of the experiments i.e., conflict resolution by ACRA and HyCRA, mechanism used in ACRA, appropriate selection of ACRA and HyCRA, and time to complete resolution. Along with the users’ satisfaction scores, the graph also provides the standard deviation of each approach indicating the variability of users responses in conflict detection and resolution.

Figure 11 Overall average satisfaction of the users with different aspects of the proposed system.

Figure 11 indicates that the highest average score i.e., 6.16 is achieved by the “time taken to complete the resolution” aspect of the system in resolving the users’ conflicts. The second highest average score i.e., 5.85 is achieved by the “ACRA” approach in resolving users’ conflicts. The “appropriate selection of ACRA” achieved an average score of 5.81 and the “mechanisms used in the ACRA” to resolve users’ conflicts achieved 5.79 average satisfaction score. The lowest average scores were achieved by the satisfactions of the users with the “HyCRA” and the “appropriate selection of HyCRA”, which achieved average satisfaction scores of 5.55, 5.51 respectively.

From the results presented in Fig. 11, it can be concluded that in the proposed system, the users preferred ACRA over HyCRA. This might be due to the fact that HyCRA is the composition of the MeCRA and the ACRA that involve users directly in the resolution of the conflicts. In contrast, the previous research efforts in literature, which use both resolution approaches in resolving multi-user conflicts (Otto et al., 2006; Shin, Dey & Woo, 2008, 2010; Wang et al., 2010; Shin & Woo, 2009a, 2009b; Shin, Dey & Woo, 2010; Shin & Woo, 2005) indicate that users prefer the MeCRA over the ACRA in resolving multi-user conflicts. The reason might be that previous research efforts never considered different special cases of users, which regularly occur in smart environments. Without considering users special cases, the system always chooses MeCRA approach to resolve conflicts.

Users’ preferences change as they encounter a special case, therefore, each user has a unique set of preferences for their special cases (other than normal). In the case of a special case user, the system’s MeCRA approach to conflict resolution is not sustained and ends once the system discovers any special case user. In this situation, the mediation is terminated, allowing the system to naturally adapt to the unique user preferences. Let us take two users as an example to demonstrate this situation. One of them has a medical condition that causes her to dislike frigid temperatures. The other user has no unique circumstances and prefers a cool environment. Because the special case user cannot compromise on the average value of the cold environment, and family members care for one another, they will give up their rights to use the application for the special case user, and allow the application to adapt to her preferences for a specific time, as long as she has that special case. After the special case user has recovered from her special case, the standard functioning method will be employed, and her preferences will be altered in accordance with his normal routine. When the same user recovers, she will be able to compromise on her preferences for the other home members who have the unusual case. This will assist the family to live in more harmonious situations since it will give the family members the impression that the other members care about them. The proposed system adopted the same situation. When a special case user was found during conflicts resolution, the other users gave up their rights to use applications for that special case user, mediation terminated, and the system automatically modified itself to the special case user’s preferences. This condition persisted until the special case user returned to her regular routine after recovering from her special case. During mediation, automatically adoption of the preferences of the special case users minimises the users involvement during mediation, and it is one of the contribution of this research. Also when a user is not interested in using an application, there is no need to involve their preferences in detecting and resolving the conflict. Our system does the same and in case of the non interested user in using certain application, the system automatically removes their preferences from the conflict detection and resolution. This is another important contribution of the presented work.

As for the conflict determination approach, Fig. 11 also shows that the users are satisfied with approach determination structure for selecting the the ACRA and the MeCRA for resolving the conflicts, their average satisfaction scores are 5.81 and 5.51 respectively.

Age wise group investigation and analysis of the obtained results was also performed. According to the age of the participants, the results are divided into two groups i.e., 25 years or below and 26 years or above. Figure 12 presents the age wise group analysis of the overall average satisfaction of the users about different aspects of the experiments. Results presented in Fig. 12 show that younger people who are aged 25 years or below give higher satisfaction for all the aspects of the system, except “HyCRA” and “appropriate selection of HyCRA”. It might be because the younger participants lived and were raised in the era after Mark Weiser’s vision of pervasive computing (Weiser, 1991), which advocates fulfillment of user tasks with no or a minimal distraction.

Figure 12 Age group wise analysis of the overall average satisfaction of the users.

Figure 13 is based on data from the first POTQ question, and it depicts users’ satisfaction with distributing priorities to them based on their role in the family (i.e., the parents will have higher priority than the children, and the elder son/daughter will have higher priority than the younger son/daughter). In case there are two or more users having the same priority (i.e., two brothers/sisters having the same age), the system will let the first user come to the environment having the same priority to control the appliances as the first come first serve (FIFO) scheme. In case of multiple users having different priorities, the system will select the highest priority user and adapt according to her preferences. These two schemes will be used to resolve the multi-user conflicts only if there is no any special case user from the involved users. Results shown in Fig. 13 indicate that 92% of the participants selected above the score three with only 2% giving the score one. Total of 4% gave it the score two, and 2% gave it the score three. A total of 4% of the participants gave it the score four, while 14% of the participants gave it the score five, and 23% gave it the score six. A total of 51% selected the highest and most satisfactory score seven, which is the most satisfactory score for this approach, in case there is no any user from the involved users having any special case. The overall average satisfaction score is 5.97.

Figure 13 Users’ satisfaction with distributing the priorities and the priority wise algorithm.

Figure 14 is drawn from the data obtained from the second question of POTQ. It shows how users rated different approaches used in the conflict resolution separately. Users were asked to rate different approaches used in the proposed system according to their opinion with different conflicting situations that may occur in the home environment. It was intended to investigate, which is the most preferred approach for the users to resolve multi-user conflicts in the context aware home environment. The proposed system considered and discussed different conflicting situations and resolved them appropriately with different approaches. The rating was to conclude, which of the resolution approaches was the most favorable to the users. The users gave different rating and in some cases the same rating for different resolution approaches.

Figure 14 Users’ ratings for the different approaches used in the proposed system.

Results shown in Fig. 14 demonstrated that the automated technique (i.e., ACRA) was the most preferred by the users. It is because, using this method, the system automatically resolved conflicts without involving users in the resolution procedures based on information obtained from their profiles. A total of 58% of the participants rated the automatic resolution as the most favorable approach, 29% of the participants rated it as average or neutral, and only 13% of the participants rated it as not favorable approach. The least favorable approach for the participants was the mediated approach with 45% of the participants rating it as the most favorable approach, 36% of the participants rated it as average or neutral, and 19% of the participants rated it as not favorable approach. The Hybrid resolution approach (i.e., HyCRA) was in between with 49% of the participants rating it as the most favorable approach, 36% rated it as average or neutral, and 15% rated it as not favorable approach.

Conclusion and future directions

Context-awareness plays a central role towards fulfilling the vision of pervasive computing outlined by Mark Weiser (Weiser, 1991), there are various interesting research challenges in the field of context-awareness (Chang, 2013). Among other research challenges in context-awareness, an issue of user conflicts in context-aware environments is very interesting and being investigated by the research community.

We identified that detecting and resolving the user conflicts in smart environments is essential. It enhances the system to support and coordinate the activities being performed by multiple users at the same time sharing the same space. Consideration of the special cases (illness, user preparing for examination, etc.) that the different users might have in context-aware environments, is very important in detecting and resolving the multi-user conflicts issue especially in the smart home environments. Despite their importance in multi-user context-aware environments, the existing works by not considering such special situations do not clearly exhibit a comprehensive solution for multi-user conflicts detection and resolution for the context-aware home environments as per requirement of the multi-user activities.

In this article, we have proposed and implemented a multi-user conflict detection and resolution system that addresses the above-mentioned conflicting situations with the special cases. The system is able to meet the needs of the home members even if they have different conflicting situations that may change from time to time. The evaluation results clearly show that the proposed system is usable, and the intended users are satisfied with the working of the system. We suggest that the work presented in this article can be extended in following directions: The proposed work on multi-user conflicts targets smart home environments, where only family members are the users of the environments. In this case, solution provided in the form of mechanisms and supporting infrastructure cannot be exploited in other smart environments, e.g., smart office, thereby requiring researching into an issue of multi-user conflicts in other context-aware environments.

Detection of special case conditions (e.g., illness of the user) and automatic update of detected special case conditions in the user profiles: Currently, in the proposed work the information about special case conditions of the users is manually inputted into their corresponding user profiles. We suggest the development and integration of infrastructure that will interact with body sensors (e.g., temperature sensor) or patient electronic health records (EHR) to detect special case conditions and update this information into its corresponding profile.

Supplemental Information

Supplemental Information 1 After Scenario Questionnaire RAW Data (Figs. 11 and 12).

Click here for additional data file.

Supplemental Information 2 Questionnaire.

Click here for additional data file.

Additional Information and Declarations

Competing Interests

Author Contributions

Data Availability

The authors declare that they have no competing interests.

Mahmoud Mohammad Aljawarneh conceived and designed the experiments, performed the experiments, analyzed the data, performed the computation work, prepared figures and/or tables, authored or reviewed drafts of the article, revised the manuscript conceptually, and approved the final draft.

Shahid Munir Shah analyzed the data, prepared figures and/or tables, authored or reviewed drafts of the article, revised the manuscript conceptually, and approved the final draft.

Lachhman Das Dhomeja performed the experiments, analyzed the data, authored or reviewed drafts of the article, and approved the final draft.

Yasir Arfat Malkani performed the experiments, analyzed the data, authored or reviewed drafts of the article, and approved the final draft.

Mahmoud Saleh Jawarneh analyzed the data, performed the computation work, prepared figures and/or tables, authored or reviewed drafts of the article, and approved the final draft.

The following information was supplied regarding data availability:

The data is available at Zenodo: Dr Mahmoud Aljawarneh, & Dr-mahmoud Jawarneh (2023). Dr Mahmoud Aljawarneh/Conflict Manager Paper: RawData (PEERJ). Zenodo. https://doi.org/10.5281/zenodo.7863589.

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
