# Peer review of "Multi-user conflict resolution mechanisms for smart home environments"

_PeerJ Computer Science, doi:10.7717/peerj-cs.1443_

## Round 0.1 · original submission · Major Revisions

Dear authors, both reviewers recommend a more precise explanation of the purpose of the work and a more straightforward structure, for example, in an IMRAD style. Please also ensure diligent proofreading before resubmission.

·

Basic reporting

Raw data not shared
Hypothesis to be explicitly mentioned

Experimental design

Kindly explain the research design explicitly
Also research question is missing
Mention the study design in the methodology
However the manuscript is written properly, it will advisable if it follows IMRAD format

Validity of the findings

Suggestion to use inferential statistics with some test of significance like t test

Additional comments

The whole process is clearly mentioned
Result section can be improved by performing some more statistics

Reviewer 2 ·

Basic reporting

The paper presents a Multi-user conflict resolution mechanism for smart home environments which takes into account users’ special cases like illness, guest visits, exams etc. for determining a resolution algorithm and an approach to be applied to detect and resolve the users’ conflicts. These users’ conflicts are resolved using automatic, mediated as well as mixed resolution approaches.
- There are A LOT of typos and weird expressions
- The paper is full of English writing errors. Some sentences are impossible to understand. A very
complete revision and correction of this aspect is essential for publication.

- Line 318: why the use of Italic???

- Fig 5 is not readable.

- Please check your references carefully

Experimental design

- The experimentation presented is too long and it does not seem to be very important for validation of the proposal. It is useful only for an illustrative purpose. I highly recommend to authors to make a motivated experimentation

Validity of the findings

- The paper starts by introducing what it seems to be the problem of Multi-user conflict. While this is a real problem, the introduction contains a lot of structural and semantical errors that make difficult to read. First, the introduction is full of disconnected phrases and gratuitous statements.
- A comparison of existing research must be provided in a table

- As such the paper presents the 'what's' of the work. To improve it, I would suggest some greater justification of why Multi-user conflict resolution mechanism for smart home is important

- It would be interesting that the authors focus on what could provide their approach in solving problems currently addressed.

Additional comments

no comment

---

## Round 0.2 · accepted · Accept

In some parts, the manuscript could be more concise; For example, in the conclusion: "an issue of user conflicts
482 in context-aware environments is very interesting and being investigated by the research community." This is a bit wordy.
However, the work is a valuable contribution worth publication.

·

Basic reporting

Overall writing is approved
and all suggestions addressed
Raw data shared

Experimental design

The design is corrected and well explained now
Flow chart provided makes the study design understable.

Validity of the findings

Results are improvised
Proper explaination provided
Discussion is appropiate

Reviewer 2 ·

Basic reporting

The article is written in English and used clear, unambiguous, technically correct text. The article is conform to professional standards of courtesy and expression.

Experimental design

the article includes all results relevant to the hypothesis
the conclusion is appropriately stated

Validity of the findings

Methods are well described with sufficient information